# The intersection of food insecurity and child health: Implications for policy and practice in the Bronx

**Lily Lee**[1,2]*, **Fatoumata Diallo**[1], **Lauren J. Shiman**[1]

**1** Bureau of Bronx Neighborhood Health, Center for Health Equity and Community Wellness, New York City Department of Health and Mental Hygiene, Bronx, New York, United States of America, **2** Department of Pediatrics, Children's Hospital at Montefiore, Bronx, New York, United States of America

* llee12@health.nyc.gov

## Abstract

### Background

Food insecurity, a significant stressor for many US families and children, is strongly associated with poverty and increases the risk of adverse physical and mental health outcomes. Communities like the Bronx, disproportionately affected by the COVID-19 pandemic due to job losses and increased reliance on emergency food resources, experience heightened food insecurity and face increased risk of various health conditions, such as asthma. This study aims to investigate the relationship between household food insecurity risk and child health outcomes in the Bronx.

### Methods

A cross-sectional analysis was conducted using data from the 2021 NYC KIDS, a population-based sample of households with children aged 1–13 years. A total of 1646 households residing in the Bronx were included in the analysis. Descriptive statistics and weighted multivariable logistic regression models were employed to examine the association between food insecurity risk and specified childhood health outcomes.

### Results

The study revealed a high prevalence of food insecurity risk among children in the Bronx, with over half residing in food-insecure households (58.5%). Household food insecurity risk was significantly associated with increased odds of overweight or obesity (adjusted OR: 1.58, 95% CI:1.01–2.47), as well as mental health disorders such as anxiety, depression, adjustment disorders, and learning disorders.

**Data availability statement:** All data is held in a public repository: https://data.cityofnewyork.us/Health/2017-2021-NYC-KIDS-Survey/u7vp-i37z/about_data.

**Funding:** The author(s) received no specific funding for this work.

**Competing interests:** The authors have declared that no competing interests exist.

## Discussion

These findings underscore the critical need for increased investment in food assistance programs within the Bronx. Despite government assistance efforts, various factors exacerbate food insecurity, including policy changes, disruptions in the supply chain, and elevated food prices. Innovative approaches such as baby bonds, prescription produce programs, paid family leave, child tax credit and guaranteed basic income should be considered to address these limitations and enhance food security in the Bronx.

## Conclusion

This study provides important insights into the association between household food insecurity risk and child health outcomes in urban, low-income communities, emphasizing the need for targeted interventions to address food insecurity and promote health equity among vulnerable children.

## Introduction

Food insecurity, defined as limited, unstable, or inconsistent access to nutritionally adequate food, is a significant stressor for many households in the United States (US). Food insecurity is strongly associated with poverty [1–4]. It is a major public health problem that increases the risk of adverse physical and mental health outcomes across the lifespan [5], and disproportionately among children [6–10]. Nationally, households with children experience food insecurity at significantly higher rates than those without children (17.9% vs. 11.9%), an inequity that widened during the COVID-19 pandemic. While national rates remained relatively steady, food insecurity in families with children rose from 13.6% to 14.8%, whereas it declined in childless households (from 9.3% to 8.8%). This widening gap was multifactorial but in part due to the loss of nutrition supports for families such as universal-free school meals [11,12].

New York City (NYC) was the epicenter of the COVID-19 pandemic and disadvantaged NYC communities are still experiencing the reverberating financial and health impacts. The Bronx, a neighborhood with predominantly Black and Latino residents, was hit the hardest with the highest rates of both COVID-19 diagnoses and deaths in the city [13]. In addition to illness and death, the pandemic created a cascade of economic and social disruptions that increased food insecurity [14]. In 2022, an estimated 14.6% of NYC residents experienced food insecurity. Due to structural and systemic barriers to wealth and health, residents of the Bronx were disproportionally burdened: 19.7% of Bronx residents reported being food insecure [15].

Food insecurity is a direct result of systemic factors such as structural racism, community disinvestment, and economic inequality that disproportionately affect neighborhoods with higher populations of people of color [16], such as the Bronx. These structural inequalities along with targeted advertising from large food

corporations limit access to affordable and healthy food options and create a precarious situation for residents who face financial constraints [17,18]. In addition to experiencing elevated risk of food insecurity compared to the rest of NYC, the Bronx also experiences higher risk of many health conditions, in part a result of policy decisions that keep people of color trapped in cycles of poverty [16,19–21]. Successful reduction of the disease burden of these medical problems among socioeconomically vulnerable communities requires an understanding of health equity as socioeconomic health dynamics continue to perpetuate health inequities in the Bronx.

While extensive work has already demonstrated the associations between food insecurity and health outcomes in adult populations, it is critical to recognize that children are uniquely vulnerable to its effects. Household food insecurity has independently been associated with chronic illnesses such as asthma [22,23] and increased risk of overweight or obesity [24]. Household food insecurity is also associated with limited access to quality healthcare and increased use of urgent care services, potentially due to an increased likelihood to delay care through competing priorities, such as food and shelter [25,26]. Mental health consequences among are also well documented; the stress from food insecurity can have mental health consequences on the entire household, including anxiety, depression, adjustment disorders, and behavioral and learning challenges [8,27–31]. These effects are not simply due to inadequate nutrition, but also to the chronic stress and instability that food insecurity introduces into the home environment.

The purpose of this study was to examine how household food insecurity risk is associated with select health conditions among children in the Bronx. This study aims to fill an important gap by exploring potential associations between food insecurity risk and child health and development in a low-income and predominantly Black and Latino population to support targeted upstream systems- and policy-level changes to reduce future population-level disease burden. Further, data from this study were collected after the onset of the COVID-19 pandemic, which exacerbated food insecurity and deserves special attention as an important time period.

## Methods

### Data and variables

Data from the 2021 NYC KIDS survey was used for this analysis. This population-based sample is described in detail elsewhere [32]. Briefly, NYC KIDS is a cross-sectional online and telephone survey of households with children with an annual stratified randomized sample to establish citywide estimates. The 2021 NYC KIDS survey contained data from four subsamples which included: 1) respondents from the 2021 Community Health Survey who reported having a child 1–13 years, 2) a birth certificate sample which included mothers who gave birth in New York City from January 1, 2015 to January 1, 2020, 3) a random, automated school health record sample of children who attend or attended New York City public schools and were born between April 1, 2007 and March 1, 2017, and 4) a random sample of households within NYC from an address-based sample [32].

Approximately 7,979 households with one or more children aged 1–13 years were interviewed, of which 1,646 resided in the Bronx. In each household, one child was randomly selected to be the focus of the interview; therefore, the number of children in the household corresponds directly to the number of unique households. The respondents were parents, guardians or other family members with sufficient knowledge about a randomly selected child's health, doctor visits, and general activities, as well as family and neighborhood characteristics. Interviews were conducted in families' preferred language.

The study sample was restricted to 1,646 individuals who resided in the Bronx. Analyses were conducted using an available case approach, in which participants were excluded from a given model only if they had missing data for that model's outcome, exposure, and covariates. For example, 248 (15.1%) respondents were excluded from analyses involving individual-level exposure variables or demographic characteristics due to missing values. Only 24.1% of observations in the original dataset had missing values in at least one of the outcome variables. Missingness across covariates was minimal (<0.2%), with the exception of housing type, which had 15.1% missing values. The number and percentages

of missing values for outcome, confounding variable, and for food insecurity are presented in S1 Table. The primary exposure measure, risk for food insecurity, was assessed by the following questions, derived from the Hunger Vital Sign screening tool: 1) *"Within the past 12 months, we worried whether our food would run out before we got money to buy more. Was this…"* and 2) *"Within the past 12 months, the food we bought didn't last and we didn't have money to get more. Was this…"* Response options for both questions were *"Often true"*, *"Sometimes true"*, *"Never true"*, *"Don't know"*, and *"Missing"*. The latter two responses were recoded to missing. Participants who responded *"Often true"* or *"Sometimes true"* to either question were identified as being at risk of having food insecurity.

Outcome variables were health outcomes affecting children that the authors hypothesized could be associated with food insecurity based on existing literature: asthma, overweight/obesity, depression, anxiety, adjustment disorder, attention-deficit disorder (ADD)/attention-deficit/hyperactivity disorder (ADHD), behavioral or conduct problems, and learning disorders or problems. For asthma, participants were asked if the child has had an episode of asthma or an asthma attack within the past 12 months. Overweight/obesity, depression, anxiety, ADD/ADHD, behavioral or conduct problems, and learning disorders or problems were assessed by asking respondents if a doctor or other healthcare provider has ever told them that the child has the condition in question.

Participants self-reported individual-level sociodemographic information on age, race/ethnicity, insurance status, housing type, parental employment, household poverty level, child's nativity, and birth within NYC. Child age, child race, and household poverty level were considered as potential confounders based on a directed acyclic graph (DAG) completed by the analysis team a priori.

### Statistical analysis

Descriptive statistics were calculated. Bivariate and multivariable logistic regression models were performed to examine the association between food insecurity risk and each specified childhood health outcome. Adjusted models included child age, child race/ethnicity, household poverty level, insurance type, parent highest level of education, parental employment status, and receipt of public assistance.

A significance level of 0.05 was used to determine statistical significance. Analyses were conducted separately by a primary and secondary data analyst (FD and LL) who subsequently met to discuss discrepancies in coding and come to a consensus. All analyses were conducted using SAS Enterprise Guide 7.0 and incorporated weights and survey design parameters. Survey weights were applied to account for the complex, multi-frame sampling design of the NYC KIDS survey and to ensure estimates are representatives of all NYC children aged 1–13 years. The weights adjust for unequal probabilities of selection, non-response, and demographic differences between the sample and the NYC population. For example, in households with both younger (1–4 years) and older (5–13 years) children, younger children had a slightly higher probability of selection; the weighting procedure corrected for this to maintain representativeness. Although some subgroups, such as older children, were somewhat underrepresented in the unweighted sample, post-stratification and raking procedures calibrated the final weights to population totals from the 5-year 2015–2019 American Community Survey to minimize potential bias in estimates. Analyses used secondary and de-identified data. NYC Health Department's Institutional Review Board deemed the secondary use of the 2021 NYC KIDS survey as exempt human subjects research. The SAS analysis code used for this study is available from the corresponding author upon request.

### Results

Descriptive characteristics of participants at the household/parent and child level, by food insecurity status, are shown in Table 1. In total, 58.5% of children in the study sample were living in households at risk of food insecurity. Most children were between 5–13 years of age (68.2%), Latino (62.0%) or Black (26.9%), insured through Medicaid or Child Health Plus insurance (77.6%), living in households with income less than 200% of the Federal Poverty Level (73.0%), born in the US (90.7%), and born in NYC (85.5%).

**Table 1.** Demographic characteristics among children in the Bronx aged 1-13 years, 2021.

| | All Bronx | | | | | Households at risk of food insecurity | | | Households not at risk of food insecurity | | |
|---|---|---|---|---|---|---|---|---|---|---|---|
| | Unweighted | | Weighted | | | Weighted | | | Weighted | | |
| | # | % | # | % | 95% CI | # | % | 95% CI | # | % | 95% CI |
| Overall | 1,646 | 100.0 | 249,000 | 100.0 | (-) | 145,000 | 58.5 | (55.3-61.7) | 104,000 | 41.5 | (38.3-44.7) |
| **Age group** | | | | | | | | | | | |
| 1–4 years | 872 | 53.0 | 79,000 | 31.8 | (29.4-34.2) | 45,000 | 31.2 | (28.0-34.6) | 34,000 | 32.7 | (28.9-36.7) |
| 5–13 years | 774 | 47.0 | 170,000 | 68.2 | (65.8-70.6) | 100,000 | 68.8 | (65.4-72.0) | 70,000 | 67.3 | (63.3-71.1) |
| **CHILD's Race/Ethnicity** | | | | | | | | | | | |
| Asian/PI, non-Latino | 63 | 3.8 | 8,000 | 3.3 | (2.3-4.5) | 4,000 | 2.5 | (1.5-4.2) | 4,000 | 4.3 | (2.8-6.6) |
| Black, non-Latino | 348 | 21.1 | 67,000 | 26.9 | (23.9-30.0) | 34,000 | 23.7 | (20.0-27.7) | 33,000 | 30.9 | (26.3-36.0) |
| Latino | 1086 | 66.0 | 154,000 | 62.0 | (58.8-65.2) | 101,000 | 69.5 | (65.3-73.4) | 54,000 | 51.8 | (46.7-56.7) |
| Other/Multi-racial, non-Latino | 69 | 4.2 | 6,000 | 2.3 | (1.7-3.1) | 3,000 | 1.8 | (1.2-2.9) | 3,000 | 3.0 | (1.9-4.6) |
| White, non-Latino | 80 | 4.9 | 14,000 | 5.6 | (4.3-7.2) | 4,000 | 2.4 | (1.5-4.1) | 10,000 | 10.0 | (7.3-13.5) |
| **Insurance Type** | | | | | | | | | | | |
| Medicaid or Child Health Plus | 1240 | 75.3 | 193,000 | 77.6 | (74.9-80.1) | 121,000 | 83.4 | (80.1-86.3) | 72,000 | 69.4 | (64.7-73.6) |
| Other insurance coverage | 395 | 24.0 | 53,000 | 21.4 | (19.0-24.1) | 22,000 | 15.2 | (12.5-18.4) | 31,000 | 30.1 | (25.9-34.7) |
| No insurance coverage | 10 | 0.6 | 3,000 | 1.0* | (0.5-2.2) | 2,000 | 1.4* | (0.6-3.2) | 1,000 | 0.5* | (0.1-2.7) |
| **Housing Type** | | | | | | | | | | | |
| A public housing resident living in a building owned by the New York City Housing Authority | 243 | 14.8 | 39,000 | 18.6 | (16.0-21.6) | 25,000 | 18.6 | (15.3-22.4) | 15,000 | 18.6 | (14.4-23.7) |
| Part of a household that receives rental assistance such as Section 8 or any other rental assistance program | 271 | 16.5 | 50,000 | 23.8 | (20.8-27.1) | 31,000 | 23.9 | (20.0-28.1) | 18,000 | 24 | (19.2-29.5) |
| Part of a household living in a rent-controlled or rent-stabilized home | 245 | 14.9 | 32,000 | 15.3 | (13.1-17.8) | 20,000 | 15.2 | (12.4-18.5) | 12,000 | 15.1 | (11.7-19.1) |
| None of these | 639 | 38.8 | 89,000 | 42.3 | (38.8-45.8) | 56,000 | 42.4 | (38.0-46.9) | 33,000 | 42.4 | (36.8-48.2) |
| **Parent Education Level** | | | | | | | | | | | |
| Less than High School | 229 | 13.9 | 30,000 | 12.1 | (10.2-14.2) | 23,000 | 16.0 | (13.2-19.2) | 7,000 | 6.6 | (4.6-9.5) |
| High School Grad | 384 | 23.4 | 70,000 | 28.0 | (25.1-31.1) | 46,000 | 32.0 | (28.0-36.2) | 23,000 | 22.3 | (18.3-27.0) |
| Some College/Technical School | 487 | 29.6 | 87,000 | 35.2 | (32.0-38.4) | 53,000 | 36.2 | (32.1-40.5) | 35,000 | 33.6 | (28.9-38.6) |
| College Graduate | 544 | 33.1 | 62,000 | 24.8 | (22.2-27.5) | 23,000 | 15.9 | (13.2-19.0) | 39,000 | 37.4 | (32.9-42.2) |
| **Parent Employment** | | | | | | | | | | | |
| Yes | 876 | 53.2 | 130,000 | 52.5 | (49.2-55.7) | 68,000 | 47.2 | (42.9-51.6) | 62,000 | 59.8 | (54.8-64.6) |
| No | 585 | 35.5 | 88,000 | 35.3 | (32.3-38.5) | 57,000 | 39.5 | (35.3-43.8) | 31,000 | 29.4 | (25.1-34.0) |
| Did not work past 30 days | 181 | 11.0 | 30,000 | 12.2 | (10.2-14.7) | 19,000 | 13.3 | (10.6-16.6) | 11,000 | 10.8 | (7.9-14.8) |
| **Household Poverty Level** | | | | | | | | | | | |
| <200% FPL | 1,149 | 69.8 | 182,000 | 73.0 | (70.1-75.7) | 117,000 | 80.5 | (76.9-83.6) | 65,000 | 62.2 | (57.4-66.7) |
| 200% FPL | 497 | 30.2 | 67,000 | 27.0 | (24.3-29.9) | 28,000 | 19.5 | (16.4-23.1) | 39,000 | 37.8 | (33.3-42.6) |
| **CHILD's Nativity** | | | | | | | | | | | |
| Born in USA^ | 1,581 | 96.1 | 226,000 | 90.7 | (88.0-92.9) | 133,000 | 91.4 | (87.7-94.0) | 93,000 | 89.7 | (85.1-93.0) |
| Born outside of USA | 65 | 4.0 | 23,000 | 9.3 | (7.1-12.0) | 13,000 | 8.6 | (6.0-12.3) | 11,000 | 10.3 | (7.0-14.9) |
| **NYC born** | | | | | | | | | | | |
| Yes | 1,519 | 92.3 | 213,000 | 85.5 | (82.5-88.0) | 126,000 | 86.7 | (82.7-89.9) | 87,000 | 83.6 | (78.7-87.5) |
| No | 126 | 7.7 | 36,000 | 14.5 | (12.0-17.5) | 19,000 | 13.3 | (10.1-17.3) | 17,000 | 16.4 | (12.5-21.3) |

*(Continued)*

**Table 1.** (Continued)

| | All Bronx | | | | | Households at risk of food insecurity | | | Households not at risk of food insecurity | | |
|---|---|---|---|---|---|---|---|---|---|---|---|
| | Unweighted | | Weighted | | | Weighted | | | Weighted | | |
| | # | % | # | % | 95% CI | # | % | 95% CI | # | % | 95% CI |
| **Receipt of public assistance in past 12 months** | | | | | | | | | | | |
| Yes | 1,002 | 61.1 | 153,000 | 61.6 | (58.4-64.7) | 100,000 | 69.4 | (65.3-73.3) | 52,000 | 50.6 | (45.7-55.6) |
| No | 638 | 38.9 | 95,000 | 38.4 | (35.3-41.6) | 44,000 | 30.6 | (26.7-34.7) | 51,000 | 49.4 | (44.4-54.3) |

Population estimates are rounded to the nearest thousand.

*Estimate should be interpreted with caution. Estimate's Relative Standard Error (a measure of estimate precision) is greater than 30%, or the 95% Confidence Interval half-width is greater than 10 or the sample size is too small, making the estimate potentially unreliable.

^Includes United States or US territories.

Source: New York City KIDS (NYC KIDS) Survey, 2021.

2021 NYC KIDS data are weighted to the population of children ages 1–13 as per the 2015–2019 American Community Survey. Data are not age adjusted.

PI = Pacific Islander; FPL = federal poverty level.

Several health outcomes were associated with household food insecurity risk, even when adjusting for child's age group, child's race/ethnicity, and household poverty level (Table 2).

Food insecurity risk was associated with 58% increase in the odds of having overweight or obesity (OR: 1.58, 95% CI: 1.01–2.47) in the adjusted model. Similarly, household food insecurity risk was associated with odds of being diagnosed with mental health disorders, including increased odds of anxiety (OR:2.68, 95% CI:1.28–5.60), depression (OR: 7.26, 95% CI:1.73–30.62), and adjustment disorders (OR:14.2, 95% CI:3.38–59.94). Household food insecurity risk was also associated with learning disorders (OR:1.7, 95% CI: 0.95–3.26). Asthma, ADD/ADHD, and behavioral problems were found to not be associated with household food insecurity risk in either crude or adjusted analyses.

**Table 2.** Univariate and multivariate logistic regression analyses of health outcomes among children in the Bronx aged 1-13 years, 2021.

| | Asthma (N=1645) | | Obesity/ Overweight (N=1646) | | Depression (N=1249) | | Anxiety (N=1249) | | Adjustment Disorders (N=1247) | | ADD/ADHD (N=1248) | | Behavioral Problems (N=1248) | | Learning Disorders (N=1249) | |
|---|---|---|---|---|---|---|---|---|---|---|---|---|---|---|---|---|
| | OR | 95% CI | OR | 95% CI | OR | 95% CI | OR | 95% CI | OR | 95% CI | OR | 95% CI | OR | 95% CI | OR | 95% CI |
| **Unadjusted model** | 1.16 | (0.81-1.68) | 1.78 | (1.18-2.70) | 7.21 | (1.52-34.15) | 3.22 | (1.61-6.43) | 17.24 | (4.24-70.08) | 1.51 | (0.86-2.66) | 1.98 | (0.94-4.19) | 2.07 | (1.12-3.83) |
| **Adjusted model^** | 1.03 | (0.69-1.54) | 1.58 | (1.01-2.47) | 7.28 | (1.73-30.62) | 2.68 | (1.28-5.60) | 14.24 | (3.38-59.94) | 1.47 | (0.82-2.65) | 1.86 | (0.80-4.34) | 1.76 | (0.95-3.26) |

*Statistically significant p<0.05.

^Adjusted for age group, child's race/ethnicity, household poverty level, insurance type, parent education level, parent employment, and receipt of public assistance.

N indicates the number of respondents with complete data for each health outcome. Both unadjusted and adjusted models were estimated using the same subset of respondents for each outcome; no additional exclusions occurred due to missing data on confounders.

Source: New York City KIDS (NYC KIDS) Survey, 2021.

2021 NYC KIDS data are weighted to the population of children ages 1-13 as per the 2015-2019 American Community Survey.

ADD = attention-deficit disorders; ADHD = attention-deficit/hyperactivity disorder.

## Discussion

This study sought to explore the complex relationship between household food insecurity risk and child health outcomes. Consistent with previous research [33–36], this study revealed a high prevalence of food insecurity among children in the Bronx, with over half residing in households at risk, more than double the national rate [37]. This underscores the persistent and pervasive nature of food insecurity, particularly in marginalized communities, despite ongoing efforts to mitigate its impact. This study also demonstrated significant association between household food insecurity and some adverse health outcomes among children. Notably, children living households with high risk of food insecurity were more likely to experience overweight or obesity, which aligns with existing literature highlighting the role of food insecurity in shaping dietary behaviors and contributing to inequities in obesity prevalence among socioeconomically disadvantaged groups [38–43]. This relationship may be driven by irregular meal patterns, limited access to nutritious foods, and higher consumption of energy-dense, low-cost foods, all of which have been shown to shape dietary behaviors and contribute to inequities in obesity prevalence, especially among socioeconomically disadvantaged groups.{Caspi, 2017 #72;Feinberg, 2008 #73} As a result of that association, food insecurity can also increase vulnerability to a myriad of other health conditions associated with obesity, such as cardiovascular diseases and type 2 diabetes [44,45].

Household food insecurity risk was also associated with higher odds of mental health disorders, including anxiety, depression, adjustment, and learning disorders. However, the study did not find associations between household food insecurity risk and conditions like asthma, ADD/ADHD, and behavioral problems.

These findings confirm that there is need for increased local, state, and federal investment into food assistance and poverty alleviation, particularly in the Bronx, a community adversely impacted by child food insecurity and multiple child comorbid conditions. Despite government efforts aimed at addressing food insecurity in the Bronx, various factors have exacerbated the issue including the removal of the Child Tax Credit in 2021, disruptions in the supply chain, elevated food prices, and limited eligibility to access SNAP benefits (commonly known as food stamps) and the Woman, Infants and Children (WIC) program. It is evident that these measures are inadequate to meet the current level of need. To address these limitations and enhance food security in the Bronx, policymakers should consider several evidence-informed policy approaches. Baby bonds, government-funded trust accounts for children, are designed to reduce long-term economic wealth gaps by providing financial resources for families with children [46,47]. Prescription produce programs, increasingly implemented in clinical settings, allow healthcare providers to "prescribe" fruits and vegetables to food-insecure patients and have been shown to improve diet quality and reduce food insecurity. [48,49]. Guaranteed basic income programs have demonstrated improved financial stability and food access among low-income households [50]. Additionally, the New York State Comptroller's Office advocates for the renewal of the Federal Child Tax Credit Expansion as an additional measure to combat poverty and food insecurity. Research indicates that recipients of the expanded federal Child Tax Credit experienced notable reductions in food insecurity compared to non-recipients [51–53]. These interventions, many of which are already implemented in other municipalities, represent feasible and impactful strategies to improve food security and health outcomes in communities like the Bronx.

While this study contributes valuable insights into the relationship between household food insecurity risk and child health outcomes, the findings should be interpreted in the context of several limitations. The cross-sectional nature of the study limits the ability to establish causality or temporality in the observed associations. The NYC KIDS survey assesses risk of food insecurity using the Hunger Vital Sign, a validated 2-item screening tool for food insecurity, rather than the full 18-item US Department of Agriculture (USDA)-Economic Research Service food security module. The Hunger Vital Sign measures marginal food insecurity, which is a broader, less severe category of food insecurity. This measure identifies households that have experienced problems or anxiety about accessing adequate food, but whose actual food intake and diet quality may not yet have been substantially reduced. Consequentially, the outcome in this study is best interpreted as marginal food insecurity, rather than food insecurity as defined by the full USDA criteria.

Additionally, certain health conditions may be underreported due to reliance on self-reported data and many conditions remain undiagnosed, especially in historically disadvantaged communities [54].

Furthermore, the small number of cases for some outcomes, such as depression and adjustment disorders, reduced the statistical power of the analysis and led to wide confidence intervals. Complete case analysis was used due to relatively minimal missingness of key variables, but it is possible that those excluded due to missing data were different from those included in the analyses. Additionally, while a DAG was used to determine the appropriate variables to include in adjusted models, it is possible that confounding bias is introduced by unmeasured variables such as parental mental health, family dynamics, and neighborhood characteristics. The study did not assess the potential mediating or moderating effects of social support, access to healthcare, or community resources on the relationship between food insecurity and child health outcomes. Understanding these factors could provide valuable insights into potential intervention targets to mitigate the adverse effects of food insecurity on child health. Future studies would benefit from exploring these additional considerations. Building on existing knowledge about the causes of food insecurity, further efforts should focus on implementing and scaling interventions that increase social support, enhance access to healthcare, and strengthen community resources to minimize health inequities. These actions, grounded in a robust theoretical framework, have the potential to reduce food insecurity in the short term and contribute to better health outcomes in the long term.

Despite these limitations, this study provides important findings about the association between household food security risk and select child health outcomes that are generalizable to urban, low-income communities of color. In addition, this study highlights the need for targeted interventions and policies that directly address food insecurity as a critical factor affecting the well-being of children and families. To our knowledge, this is the first study exploring these associations in a low-income and predominantly Black and Brown population in the US, where particular attention should be paid to the potential consequences of food insecurity in childhood. Efforts to improve food access, promote nutrition education, and strengthen social safety nets are critical steps toward creating healthier and more equitable communities for all children.

## Supporting information

**S1 Table. Missing values for outcome, confounding variables, and food insecurity among children in the Bronx, 2021.**
(DOCX)

## Author contributions

**Conceptualization:** Fatoumata Diallo, Lauren J. Shiman.

**Formal analysis:** Lily Lee, Fatoumata Diallo.

**Methodology:** Fatoumata Diallo.

**Supervision:** Lauren J. Shiman.

**Writing – original draft:** Lily Lee, Lauren J. Shiman.

**Writing – review & editing:** Lily Lee, Fatoumata Diallo, Lauren J. Shiman.

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
