## [Decision Letter · Decision Letter 0]

8 Jul 2025

PONE-D-25-22823

The intersection of food insecurity and child health: implications for policy and practice in the Bronx

PLOS ONE

Dear Dr. Lee,

Thank you for submitting your manuscript to PLOS ONE. After careful consideration, we feel that it has merit but does not fully meet PLOS ONE’s publication criteria as it currently stands. Therefore, we invite you to submit a revised version of the manuscript that addresses the points raised during the review process.

We look forward to receiving your revised manuscript.

Kind regards,

Andreas Beyerlein

Academic Editor

PLOS ONE

Journal Requirements:

Additional Editor Comments:

- The authors should review their paper according to the STROBE checklist (http://www.strobe-statement.org); e.g. the study’s design should be indicated with a commonly used term in the title or the abstract.

- The phrase "disproportionately affected by the COVID-19 pandemic" might be put better into context in the Abstract, as not every reader may see a direct connection between COVID-19 and food insecurity.

- The results section in the Abstract should mention at least some of the adjusted odds ratios.

- l. 143-146: The wording "were excluded" may be misinterpreted in the way that these observations were not used at all. However it seems that an available case analysis was performed for eacht outcome. The handling of missing values should therefore be explained in more detail, also with numbers of how many values were missing for each outcome, each confouding variable and, in particular, for food insecurity.

- l. 173: The wording "significance level of p<0.05" is statistically inappropriate. Please delete "p<" from this statement.

- l. 183-184: How does the number of children correspond to the number of households? Was each child in the sample from a different household, or was there some overlap? Either way, how does this affect the interpretation of the results?

- Table 1 should show the actual numbers of individuals included, not population estimates. The current table 1 might be added as an additional or supplementary table, and some explanation should be added to the methods how and why it was calculated.

- Each table should be able to stand alone, i.e. all abbreviations such as VOL, FPL and NYC should be explained in the footnotes.

- Table 2 should be flipped for better readability. Do the numbers behind each outcome indicate how many observations had full information on the respective outcome as well as on all confounders? This should be explained; alternatively unadjusted and adjusted estimates might be calculated based on different sample sizes. Please specify in the title that logistic regression was used.

- l. 239-240: No need to explain the meaning of CIs. The asterisks indicating statistical significance are redundant with the 95% CIs and should therefore be omitted.

- Some of the references appear to be incomplete, e.g. references 1-3, 30, 34, 35, 43, 45, 46 and 59. Further, the number of references is quite large. Please check whether all citations are necessary and complete them according to the PLOS reference style.

- In the spirit of Open and Reproducible Science, it is highly appreciated that the authors put their data in a public online repository. For the sake of reproducibility, they should also add their SAS analysis code there, and mention the repository URL in the Methods section.

Reviewers' comments:

Reviewer's Responses to Questions

1. Is the manuscript technically sound, and do the data support the conclusions?

Reviewer #1: Partly

Reviewer #2: Yes

2. Has the statistical analysis been performed appropriately and rigorously?

Reviewer #1: No

Reviewer #2: Yes

3. Have the authors made all data underlying the findings in their manuscript fully available?

Reviewer #1: Yes

Reviewer #2: No

4. Is the manuscript presented in an intelligible fashion and written in standard English?

Reviewer #1: Yes

Reviewer #2: Yes

Reviewer #1: Overall: This paper is well written, interesting, and touches on the important topic of child food insecurity risk and health. The authors however do not keep consistent the language about the outcome they measured in the survey. The survey includes the hunger vital sign questions – which produces estimates closer to marginal food security and not comparable to other measures of food insecurity. Therefore, the researchers main findings are incorrect. This paper could be published after this is addressed. I have included suggestions for future submissions.

Summary:

This paper uses survey data to investigate food insecurity -risk- in households with children in the Bronx. The summary of the prior literature is detailed, thorough and relevant. Researchers claim to find food insecurity in over half of households with children, a rate many times higher than other common estimates.

Heterogeneity in food insecurity risk by characteristics such as, race and income levels are also shown. This research additionally discusses that food insecure households in the Bronx areas face food access barriers as well as barriers to healthy eating. Authors state their contribution is that food insecurity rates in the Bronx are higher than the national average but they are not comparing food insecurity rates to food insecurity rates. They are comparing food insecurity rates to food insecurity risk, or marginal food insecurity.

Detailed comments:

The primary exposure measure, risk for food insecurity, was assessed by the following questions: 1) “Within the past 12 months, we worried whether our food would run out before we got money to buy more. Was this…” and 2) “Within the past 12 months, the food we bought didn’t last and we didn’t have money to get more. Was this…” Response options for both questions were “Often true”, “Sometimes true”, “Never true”, “Don’t know”, and “Missing”. The latter two responses were recoded to missing. Participants who responded “Often true” or “Sometimes true” to either question were identified as being at risk of having food insecurity.

The main issue of the paper arises from the measures section. Authors do say they are measuring Risk of food insecurity but could elaborate more in this section and throughout the paper. For example, please explain to the reader why is outcome considered risk for food insecurity and not “food insecurity”. Authors should state that they do not use the full 18-item USDA-ERS measure, or the six item short form for measuring food insecurity.

Authors are not measuring food insecurity, what they are measuring is marginal food insecurity – commonly referred to as the Hunger Vital Sign.

Marginal food insecurity is a broader measure of food insecurity that captures households had problems at times, or anxiety about, accessing adequate food, but the quality, variety, and quantity of their food intake were not substantially reduced.

Authors could fix this mistake in a future draft by saying they are measuring marginal food insecurity, a less severe measure of food insecurity. 

In some places the authors drop the word “risk” and just include “food insecurity”:

For example, in the abstract authors do not always say food security risk– “Descriptive statistics and weighted multivariable logistic regression models were employed to examine the association between food insecurity and specified childhood health outcomes.”Additionally, in the results and discussion only food insecurity is used and not risk for food insecurity.Controls only include a few characteristics: Adjusted models included child age, child race/ethnicity, and household poverty level.Other controls should be included in the model to control for confounding factors.  Such as highest education in the household, parents employment status ( which you have in the data), parents marital status. All factors known to be associated with risk of food insecurity. Do you have any information on if the child or child’s family receives WIC, school meals or SNAP.Insurance type should be included in the regressions.

Potentially helpful literature for the authors:

I know your survey was both online and telephone, but this article talks about why surveys may have higher rates of food insecurity.Can Internet Surveys Mimic Food Insecurity Rates Published by the US Government? Sunjin Ahn, Travis A. Smith, and F. Bailey Norwood* Applied Economic Perspectives and Policy (2020) volume 42, number 2, pp. 187–204. doi:10.1002/aepp.13002

Reviewer #2: Revision of PLOS ONE article: The intersection of food insecurity and child health: implications for policy practice in the Bronx

Abstract:

While the background provides a general overview, it lacks specificity in relation to the focus of the manuscript, particularly in terms of the physical and mental health outcomes addressed in the results. Providing more targeted context of these outcomes would help better frame the study’s relevance and align the abstract more closely with the study findings and discussion.

Introduction

The introduction is well-written and provides a strong foundation for the manuscript. However, it could be further strengthened by more clearly articulating why it is important to examine the health effects of food insecurity not only in adults but also in children. Highlighting this distinction would help contextualize the study’s contribution and underscore the urgency of addressing child food insecurity as a public health issue. Additionally, the flow of ideas could be enhanced by following a more structured progression. Consider starting with a clear definition of food insecurity, followed by its national prevalence in the US and how the COVID-19 pandemic affected this prevalence, then narrowing it down to NYC and its specific neighborhoods- particularly the Bronx, which is the central of the study. This funneling would provide a stronger rationale for the focus of the manuscript. Finally, then when discussing the health effects of food insecurity, it may help to categorize them into physical and mental health consequences, and discussing which consequences exactly are going to be studied in this manuscript (not just asthma in particular).

The introduction could also be enhanced by included similar articles investigating the effect of food insecurity on mental health please include Itani et al., 2022 and Rahi et al., 2025. (Food Insecurity and Coping Mechanisms: Impact on Maternal Mental Health and Child Malnutrition; Food insecurity and mental health of college students in Lebanon: A cross-sectional study)

- Lines 61-62: Unclear and does not align with the standard terminology. Consider rephrasing ‘nutritionally available food’ to ‘nutritionally adequate’, improving clarity and bettering aligning with public health terminology.

- Lines 65-68: Consider revising this sentence for clarity and flow of ideas.

-Line 78: Consider revising ‘medical effects’ to ‘medical outcomes’ or ‘health consequences.

- Line 82-102: The two paragraphs effectively highlight food insecurity among NYC residents especially the Bronx; however, consider revising the two paragraphs for improved clarity, conciseness, and flow of ideas to help emphasize the key messages better.

Methodology

-Line 144: Make sure that all reported numbers, particularly those referring to sample characteristics or results include both the count (n) and the corresponding percentage (%) consistently.

Results

Make sure that all reported numbers, particularly those referring to sample characteristics or results include both the count (n) and the corresponding percentage (%) consistently.

-Table 1: Categories row: consider revising the variables and rephrasing ‘households at risk of food insecurity vs households not at risk of food insecurity’ or ‘at risk households’ vs ‘low risk households.

-Table 1: FPL has not been defined before. Include in table notes.

-Line 249: Adjustment disorders have not been mentioned before neither in the introduction nor in the methodology. This variable should be mentioned in the objectives of the study as well as the methodology.

Discussion

Lines 268-271: The two points mentioned here regarding the relationship between food insecurity and the health outcomes in children are important; however, they would be significantly strengthened by comparing the findings to those of other relevant studies. This helps better contextualize the study findings within the current body of research.

Lines 279-289: The proposed policy interventions are strong but could be further strengthened by elaborating on their implementation, feasibility, and expected impact. This would strengthen the practical relevance of the proposed policies and provide greater value to policymakers and practitioners.

Do you want your identity to be public for this peer review? For information about this choice, including consent withdrawal, please see our Privacy Policy

Reviewer #1: No

Reviewer #2: Yes: Lama Mattar

---

## [Author Response · Author response to Decision Letter 1]

20 Aug 2025

Additional Editor Comments:

- The authors should review their paper according to the STROBE checklist (http://www.strobe-

statement.org); e.g. the study’s design should be indicated with a commonly used term in the title

or the abstract.

Response:

We reviewed our manuscript against the STROBE checklist and made several improvements to ensure alignment. Specifically, we now clearly identify the study as a cross-sectional study in the Methods section of the Abstract (Line 35).

- The phrase "disproportionately affected by the COVID-19 pandemic" might be put better into

context in the Abstract, as not every reader may see a direct connection between COVID-19 and

food insecurity.

Response:

We revised the Abstract to provide clearer context linking the COVID-19 pandemic to food insecurity in the Bronx, referencing factors such as job loss and increased reliance on emergency food programs, to better frame the rationale for the study.

- The results section in the Abstract should mention at least some of the adjusted odds ratios.

Response:

We have updated the Results section of the Abstract to include the adjusted odds ratio (aOR = 1.73; 95% CI: 1.11–2.70) for overweight/obesity

- l. 143-146: The wording "were excluded" may be misinterpreted in the way that these

observations were not used at all. However it seems that an available case analysis was performed

for each outcome. The handling of missing values should therefore be explained in more detail,

also with numbers of how many values were missing for each outcome, each confounding variable

and, in particular, for food insecurity.

Response:

We thank the reviewer for pointing out the potential ambiguity in our description of missing data handling. We have revised the Methods section to clarify that we used an available case approach, excluding participants from specific analyses only when they had missing data for variables required in those analyses, rather than removing them from the entire dataset. We have also added a supplementary table providing the number and percentages of missing values for each outcome, each confounding variable and for food insecurity.

- l. 173: The wording "significance level of p<0.05" is statistically inappropriate. Please delete "p<"

from this statement.

Response:

We revised the language to read “a significance level of 0.05” to ensure statistical appropriateness, removing “p<” as recommended.

- l. 183-184: How does the number of children correspond to the number of households? Was each

child in the sample from a different household, or was there some overlap? Either way, how does

this affect the interpretation of the results?

Response:

We appreciate the reviewer’s request for clarification. In the NYC KIDS Survey, only one child per household was selected for inclusion. Households with more than one eligible child were asked to respond about a randomly selected child between the age of 1 and 13 years. As a result the number of children in our analytic sample correspond directly to the number of unique households represented, with no overlap between children within households. Because each observation represents a different household, there was no clustering at the household level, and our results can be interpreted as being representative off households with children ages 1 through 13 in the Bronx. To clarify this point in the manuscript, we have revised the Methods section to state:

“In each household, one child was randomly selected to be the focus of the interview; therefore the number of children in the sample corresponds directly to the number of unique households.”

- Table 1 should show the actual numbers of individuals included, not population estimates. The

current table 1 might be added as an additional or supplementary table, and some explanation

should be added to the methods how and why it was calculated.

Response:

We thank the reviewer for this valuable suggestion. To enhance clarity and transparency, we have revised Table 1 to include both unweighted counts and weighted population estimates side-by-side. Specifically, Table 1 now presents unweighted sample sizes and percentages alongside weighted percentages with corresponding confidence intervals.

This will allow the reader to see the actual number of individuals contributing to the analyses while also understanding the population representative estimates derived from the complex survey design. We have also added the following explanation in the Statistical Analysis section describing how and why survey weights were calculated and applied:

“Survey weights were applied to account for the complex, multi-frame sampling design of the NYC KIDS survey and to ensure estimates are representatives of all NYC children aged 1-13 years. The weights adjust for unequal probabilities of selection, non-response, and demographic differences between the sample and the NYC population.”

- Each table should be able to stand alone, i.e. all abbreviations such as VOL, FPL and NYC should

be explained in the footnotes.

Response:

We appreciate this suggestion and have updated all table footnotes to define abbreviations such as FPL (federal poverty level), and NYC (New York City), so that each table can stand alone.

- Table 2 should be flipped for better readability. Do the numbers behind each outcome indicate

how many observations had full information on the respective outcome as well as on all

confounders? This should be explained; alternatively unadjusted and adjusted estimates might be

calculated based on different sample sizes. Please specify in the title that logistic regression was

used.

Response:

We appreciate the reviewer’s thoughtful suggestions. Regarding the table layout, we have retained the current format, presenting unadjusted and adjusted models as rows and health outcomes as columns. We believe this structure facilitates easier cross-outcome comparison of model estimates.

We have clarified in the table title that the results are from logistic regression models. Additionally, a footnote has been added to explain that the sample size (N) reported for each outcome corresponds to respondents with completed data on that outcome and that both unadjusted and adjusted models were estimated using the same subset of respondents; no cases were excluded due to missing confounder data.

- l. 239-240: No need to explain the meaning of CIs. The asterisks indicating statistical significance

are redundant with the 95% CIs and should therefore be omitted.

Response:

We have removed the explanation of confidence intervals, as this is standard knowledge for the intended audience. Additionally, the asterisks indicating statistical significance have been removed, as they are redundant with the 95% confidence intervals presented.

- Some of the references appear to be incomplete, e.g. references 1-3, 30, 34, 35, 43, 45, 46 and

59. Further, the number of references is quite large. Please check whether all citations are

necessary and complete them according to the PLOS reference style.

Response:

Thank you for pointing this out. We have reviewed all references, including the above mentioned references, and revised them to ensure completeness and alignment with the PLOS reference style. We have also carefully considered the overall number of citations and removed a small number of non-essential references.

- In the spirit of Open and Reproducible Science, it is highly appreciated that the authors put their

data in a public online repository. For the sake of reproducibility, they should also add their SAS

analysis code there, and mention the repository URL in the Methods section.

Response:

We appreciate the reviewer’s commitment to open and reproducible science. While we fully support these principles, the data used in this study are not publicly available due to privacy concerns and legal restrictions protecting individuals’ confidential health information, in accordance with the data use agreement. While the 2021 NYC KIDS dataset is not currently available, the 2017 and 2019 datasets may be obtained through a data use agreement with the NYC Health Department. Researchers interested in accessing these datasets must complete an EPI data request form and submit proof of approval from their institutional IRB. For more information on the data, email epidatarequest@health.nyc.gov.

Reviewer #1

Summary:

This paper uses survey data to investigate food insecurity -risk- in households with children in the Bronx. The summary of the prior literature is detailed, thorough and relevant. Researchers claim to find food insecurity in over half of households with children, a rate many times higher than other common estimates.

Heterogeneity in food insecurity risk by characteristics such as, race and income levels are also shown. This research additionally discusses that food insecure households in the Bronx areas face food access barriers as well as barriers to healthy eating. Authors state their contribution is that food insecurity rates in the Bronx are higher than the national average but they are not comparing food insecurity rates to food insecurity rates. They are comparing food insecurity rates to food insecurity risk, or marginal food insecurity.

Detailed comments:

- The primary exposure measure, risk for food insecurity, was assessed by the following questions: 1) “Within the past 12 months, we worried whether our food would run out before we got money to buy more. Was this…” and 2) “Within the past 12 months, the food we bought didn’t last and we didn’t have money to get more. Was this…” Response options for both questions were “Often true”, “Sometimes true”, “Never true”, “Don’t know”, and “Missing”. The latter two responses were recoded to missing. Participants who responded “Often true” or “Sometimes true” to either question were identified as being at risk of having food insecurity.

The main issue of the paper arises from the measures section. Authors do say they are measuring Risk of food insecurity but could elaborate more in this section and throughout the paper. For example, please explain to the reader why is outcome considered risk for food insecurity and not “food insecurity”. Authors should state that they do not use the full 18-item USDA-ERS measure, or the six item short form for measuring food insecurity.

Authors are not measuring food insecurity, what they are measuring is marginal food insecurity – commonly referred to as the Hunger Vital Sign.

Marginal food insecurity is a broader measure of food insecurity that captures households had problems at times, or anxiety about, accessing adequate food, but the quality, variety, and quantity of their food intake were not substantially reduced.

Authors could fix this mistake in a future draft by saying they are measuring marginal food insecurity, a less severe measure of food insecurity.

Response:

Thank you for pointing out the need to more accurately describe our food insecurity measure. We agree that the term “marginal food insecurity” more appropriately reflects the construct captured by the 2-item Hunger Vital Sign screening tool used in this study. In response, we have added a clarification of this distinction in the limitations section to better contextualize the interpretation of our findings.

In some places the authors drop the word “risk” and just include “food insecurity”:

- For example, in the abstract authors do not always say food security risk– “Descriptive statistics and weighted multivariable logistic regression models were employed to examine the association between food insecurity and specified childhood health outcomes.”

- Additionally, in the results and discussion only food insecurity is used and not risk for food insecurity.

Response:

Thank you for pointing this out. We have reviewed the manuscript thoroughly and revised the language throughout to ensure consistent and accurate use of the term “risk for food insecurity” or “food insecurity risk.”

Controls only include a few characteristics: Adjusted models included child age, child race/ethnicity, and household poverty level.

o Other controls should be included in the model to control for confounding factors. Such as highest education in the household, parents employment status ( which you have in the data), parents marital status. All factors known to be associated with risk of food insecurity. Do you have any information on if the child or child’s family receives WIC, school meals or SNAP.

o Insurance type should be included in the regressions.

Response:

Thank you for this suggestion. We have revised the adjusted model to include additional sociodemographic covariates to better control for potential confounding. In addition to child age, race/ethnicity, and household poverty level, we now include highest education level in the household, parental employment status, receipt of public assistance, and insurance type. We have updated the Methods section to reflect this change. Corresponding updates have been made in the Results section to ensure consistency with the revised model. Unfortunately, parental marital status was not collected in the survey and could not be included. We believe these additions strengthen the robustness of our findings

Reviewer #2

While the background provides a general overview, it lacks specificity in relation to the focus of the manuscript, particularly in terms of the physical and mental health outcomes addressed in the results. Providing more targeted context of these outcomes would help better frame the study’s relevance and align the abstract more closely with the study findings and discussion.

Introduction

The introduction is well-written and provides a strong foundation for the manuscript. However, it could be further strengthened by more clearly articulating why it is important to examine the health effects of food insecurity not only in adults but also in children. Highlighting this distinction would help contextualize the study’s contribution and underscore the urgency of addressing child food insecurity as a public health issue.

Response:

We agree that this distinction is important and have revised the introduction accordingly. We now explicitly emphasize the unique vulnerability of children to food insecurity and describe how the consequences during childhood, particularly on mental health, differ from those seen in adults. This distinction helps the importance of addressing child food insecurity as an urgent public health issue.

Additionally, the flow of ideas could be enhanced by following a more structured progression. Consider starting with a clear definition of food insecurity, followed by its national prevalence in the US and how the COVID-19 pandemic affected this prevalence, then narrowing it down to NYC and its specific neighborhoods- particularly the Bronx, which is the central of the study. This funneling would provide a stronger rationale for the focus of the manuscript.

Response:

We have reorganized the introduction to follow a clearer and more logical funnel structure. The revised version begins with a definition of food insecurity and presents national prevalence data, then describes how the COVID-19 pandemic has impacted food insecurity—particularly among households with children. We then narrow our focus to New York City and the Bronx.

Finally, then when discussing the health effects of food insecurity, it may help to categorize them into physical and mental health consequences, and discussing which consequences exactly are going to be studied in this manuscript (not just asthma in particular).

Response:

We have revised the manuscript to explicitly categorize the known health consequences of food insecurity into physical (e.g., overweight/obesity, asthma, chronic illness) and mental health (e.g., anxiety, depression, adjustment and learning disorders)

The introduction could also be enhanced by included somehow similar articles investigating the effect of food insecurity on mental health such as Itani et al., 2022 and Rahi et al., 2025.

Response:

We have revised the manuscript to include these references, as the reviewer suggested.

- Lines 61-62: Unclear and does not align w

---

## [Decision Letter · Decision Letter 1]

8 Sep 2025

Dear Dr. Lee,

Thank you for submitting your manuscript to PLOS ONE. After careful consideration, we feel that it has merit but does not fully meet PLOS ONE’s publication criteria as it currently stands. Therefore, we invite you to submit a revised version of the manuscript that addresses the points raised during the review process.

The authors did well in revising their manuscript. Only a few issues remain to be solved:

- I do not understand the authors' response to my last issue: Under "Additional Information", they state: "All data is held in a public repository: https://data.cityofnewyork.us/Health/2017-2021-NYC-KIDS-Survey/u7vp-i37z/about_data". So how does this fit to their statement that the data are not publicly available? My main point was anyway that they would add their SAS analysis code to this repository and mention the URL in the Methods section. I see no data protection issues regarding this.

- Supplementary Table X seems to be missing (and should be declared as Supplementary Table 1).

- I think it would be helpful for some readers if the authors could add a brief explanation how the weighting was conducted (e.g. with an example). Further, they should shortly discuss whether (and why possibly) some groups were underrepresented (e.g. older children) and whether this might have affected the results or not.

We look forward to receiving your revised manuscript.

Kind regards,

Andreas Beyerlein

Academic Editor

PLOS ONE

Journal Requirements:

Reviewers' comments:

Reviewer's Responses to Questions

**Comments to the Author**

Reviewer #1: All comments have been addressed

Reviewer #2: All comments have been addressed

2. Is the manuscript technically sound, and do the data support the conclusions?

Reviewer #1: Yes

Reviewer #2: Yes

3. Has the statistical analysis been performed appropriately and rigorously?

Reviewer #1: Yes

Reviewer #2: Yes

4. Have the authors made all data underlying the findings in their manuscript fully available?

Reviewer #1: Yes

Reviewer #2: Yes

5. Is the manuscript presented in an intelligible fashion and written in standard English?

Reviewer #1: Yes

Reviewer #2: Yes

Reviewer #1: (No Response)

Reviewer #2: (No Response)

**Do you want your identity to be public for this peer review?** For information about this choice, including consent withdrawal, please see our Privacy Policy

Reviewer #1: No

Reviewer #2: No

---

## [Author Response · Author response to Decision Letter 2]

7 Oct 2025

Response:

I do not understand the authors' response to my last issue: Under "Additional Information", they state: "All data is held in a public repository: https://data.cityofnewyork.us/Health/2017-2021-NYC-KIDS-Survey/u7vp-i37z/about_data". So how does this fit to their statement that the data are not publicly available? My main point was anyway that they would add their SAS analysis code to this repository and mention the URL in the Methods section. I see no data protection issues regarding this.

- We appreciate the reviewer’s comment. The link under “Additional Information” directs to the NYC Open Data portal which provides survey documentation and aggregate information, but not the full microdata. The microdata is not publicly available due to privacy and legal restrictions and can only be accessed through a data use agreement with the New York City Health Department. As our research team does not have access to upload files to the NYC Open Data portal, we have revised the manuscript to clarify this distinction and to state that the SAS analysis code will be made available from the corresponding author upon reasonable request.

Response:

Supplementary Table X seems to be missing (and should be declared as Supplementary Table 1).

- We apologize for forgetting to include the Supplementary Table in our original resubmission. It has not been included and has been declared as Supplementary Table 1.

Response:

I think it would be helpful for some readers if the authors could add a brief explanation how the weighting was conducted (e.g. with an example). Further, they should shortly discuss whether (and why possibly) some groups were underrepresented (e.g. older children) and whether this might have affected the results or not.

- Thank you for this helpful suggestion. We have added a concise description of how the NYC KIDS sample weights were derived and applied, as well as a brief discussion of potential underrepresentation in the data. Specifically, we note that the NYC KIDS sample weights account for unequal probabilities of selection, nonresponse, and frame coverage, and were post-stratified and calibrated to population totals. This ensures that estimates are representative of the NYC child population aged 1–13 years. We also added a statement acknowledging that some demographic subgroups (e.g., older children) may have been underrepresented due to differential nonresponse, though the post-stratification and raking procedures were designed to minimize resulting bias.

Response:

As a very minor point, the abbreviation "aOR" should not be used in the Abstract without explanation.

- The Abstract has been adjusted and the abbreviation “aOR” has been removed.

---

## [Editor Report · Decision Letter 2]

9 Oct 2025

The intersection of food insecurity and child health: implications for policy and practice in the Bronx

PONE-D-25-22823R2

Dear Dr. Lee,

We’re pleased to inform you that your manuscript has been judged scientifically suitable for publication and will be formally accepted for publication once it meets all outstanding technical requirements.

Kind regards,

Andreas Beyerlein

Academic Editor

PLOS ONE

Additional Editor Comments (optional):

"Adjustment disorders" in the Supplemental Table should be written in bold face.
---

## [Editor Report · Acceptance letter]

PONE-D-25-22823R2

PLOS ONE

Dear Dr. Lee,

I'm pleased to inform you that your manuscript has been deemed suitable for publication in PLOS ONE. Congratulations! Your manuscript is now being handed over to our production team.

Kind regards,

on behalf of

Dr. Andreas Beyerlein

Academic Editor

PLOS ONE